# Simultaneous Metabarcoding and Quantification of *Neocallimastigomycetes* from Environmental Samples: Insights into Community Composition and Novel Lineages

**DOI:** 10.3390/microorganisms10091749

**Published:** 2022-08-30

**Authors:** Diana Young, Akshay Joshi, Liren Huang, Bernhard Munk, Christian Wurzbacher, Noha H. Youssef, Mostafa S. Elshahed, Christina D. Moon, Katrin Ochsenreither, Gareth W. Griffith, Tony M. Callaghan, Alexander Sczyrba, Michael Lebuhn, Veronika Flad

**Affiliations:** 1Micro and Molecular Biology, Central Department for Quality Assurance and Analytics, Bavarian State Research Center for Agriculture, 85354 Freising, Germany; 2Biocatalysis, Environment and Process Technology Unit, Life Science and Facility Management, ZHAW, 8820 Wadenswil, Switzerland; 3Department of Microbiology, University of Innsbruck, A-6020 Innsbruck, Austria; 4Center for Biotechnology (CeBiTec), University of Bielefeld, 33615 Bielefeld, Germany; 5Chair of Urban Water Systems Engineering, Technical University of Munich (TUM), 85748 Garching, Germany; 6Department of Microbiology and Molecular Genetics (OSU), Oklahoma State University, Stillwater, OK 74074, USA; 7AgResearch, Grasslands Research Centre, Palmerston North 4442, New Zealand; 8Process Engineering in Life Sciences 2: Technical Biology (KIT), Karlsruhe Institute of Technology, 76131 Karlsruhe, Germany; 9Department of Life Sciences (DoLS), Aberystwyth University, Aberystwyth SY23 3DD, Wales, UK; 10Niskus Biotec Limited Co., F92 K314 Donegal, Ireland

**Keywords:** anaerobic gut fungi, *Neocallimastigomycetes*, environmental screening, Quantitative Real-Time PCR, large ribosomal subunit, barcoding, high-throughput sequencing, phylogenetic analysis

## Abstract

Anaerobic fungi from the herbivore digestive tract (*Neocallimastigomycetes*) are primary lignocellulose modifiers and hold promise for biotechnological applications. Their molecular detection is currently difficult due to the non-specificity of published primer pairs, which impairs evolutionary and ecological research with environmental samples. We developed and validated a *Neocallimastigomycetes*-specific PCR primer pair targeting the D2 region of the ribosomal large subunit suitable for screening, quantifying, and sequencing. We evaluated this primer pair in silico on sequences from all known genera, in vitro with pure cultures covering 16 of the 20 known genera, and on environmental samples with highly diverse microbiomes. The amplified region allowed phylogenetic differentiation of all known genera and most species. The amplicon is about 350 bp long, suitable for short-read high-throughput sequencing as well as qPCR assays. Sequencing of herbivore fecal samples verified the specificity of the primer pair and recovered highly diverse and so far unknown anaerobic gut fungal taxa. As the chosen barcoding region can be easily aligned and is taxonomically informative, the sequences can be used for classification and phylogenetic inferences. Several new *Neocallimastigomycetes* clades were obtained, some of which represent putative novel lineages such as a clade from feces of the rodent *Dolichotis patagonum* (mara).

## 1. Introduction

The order *Neocallimastigomycetes* is composed of strict anaerobic fungi that inhabit the digestive tracts of a wide variety of animals, mainly ruminant herbivores, feeding on a high fiber diet [1,2]. Recently, these anaerobic gut fungi (AGF) have received considerable attention not only due to their importance as initial cellulose and hemicellulose degraders in animal nutrition, but also for their potential to make use of lignocellulosic materials in diverse biotechnological applications, e.g., by transforming agriculture waste or side products into bioenergy [3,4,5]. To date, 20 genera have been described in the class *Neocallimastigomycetes*, and comparative analysis of enzymatic activities suggests qualitative and quantitative differences in lignocellulosic modification patterns and extent between various taxa [6]. Knowledge of their diversity and main enzymatic activities can be substantially expanded if novel isolates are identified and cultivated. Deciphering these activities can be a challenge, particularly if the search aims at discovering new genera/species in unexplored habitats.

The typical habitat of *Neocallimastigomycetes* is the herbivore gut but they have also been detected in the sludge of agricultural biogas plants fed with ruminant manure [3,7] and at landfill sites [8]. Dollhofer et al. [7] measured activity of AGF in digestate from a biogas plant fed with more than 30% cattle manure and even isolated a *Piromyces* sp. strain from one of the fermenters. The question remains if members of the *Neocallimastigomycetes* can survive, inhabit, and proliferate in environments other than herbivore guts; even further, if they can be found in environments that are not related to or in contact with herbivores.

AGF isolation and cultivation can still be a time-consuming and laborious endeavor. Commonly used isolation protocols can even favor some species, neglecting the detection of the whole AGF diversity in an environmental sample. Added to this, basic microscopic analysis is not suitable for environmental samples. Molecular-guided screening and community structure determination can be advantageous to speed up AGF detection, quantification, and classification, which will help to identify interesting samples and habitats for more in-depth analysis.

The internal transcribed spacer (ITS) region (the formal fungal barcode [9]) and variable regions of the large ribosomal subunit (LSU; 28S rRNA gene), both within the ribosomal RNA operon, are currently the main molecular regions to detect, identify, and classify AGF. Both regions are referred to curated databases (UNITE for ITS, and www.anaerobicfungi.org/databases/ for ITS and LSU) to classify and characterize AGF. However, the D1–D2 region of the LSU shows clear advantages over the ITS region for AGF classification: First, it exhibits low intra-strain variation, when compared to the highly variable ITS1 region. Second, it exhibits limited length variability between the AGF taxa [10,11], thus avoiding PCR amplification bias. Third, LSU does not present extended AT rich sites making alignment easier and prevent sequencing errors [12]. Furthermore, the D1-D2 region exhibits adequate variability to successfully distinguish all known taxa on the species level using phylogenetic inference methods [13,14,15]. 

Several PCR-based assays for AGF have been proposed as convenient and reliable fast screening approaches. Nevertheless, previous LSU sets of primer pairs designed for the detection and analysis of *Neocallimastigomycetes* [12] are not specific enough to exclusively detect them in environmental samples containing diverse microbiomes. The reason is that unspecific primers can bind to any complementary nucleic acid sequence and may amplify DNA of non-target phylogenetic groups. Some of the amplicons are also too long (>600 bp) to be useful in Real-Time Quantitative PCR (qPCR) and high-throughput sequencing (HTS) such as Illumina HiSeq or MiSeq platforms producing short reads. Third generation sequencing of long reads is emerging, but quantification of these sequences is still a challenge for qPCR. Even with commonly shorter sequences, for example, the amplicons from the primer pairs 28S rRNA (approx. 1570 bp, [16]), NL1–NL4 (approx. 783 bp, [17]), and GGNL1–GGNL4 (approx. 541 bp, [18]) are too long for qPCR and most short read sequencers. Recently published primers as AF-LSU (approx. 441 bp long) described by Dollhofer et al. [14], are close to the maximum length for Illumina sequencing but do not detect all recently discovered genera. The use of primers targeting other loci is not straightforward, since these would mostly be single copy genes, thus the assays would be less sensitive. In contrast, the rRNA operon is present in multiple copy numbers in a single genome. Additionally, the known occurrence of AGF horizontal gene transfer from *Bacteria* to *Neocallimastigomycetes* [19], for example of genes encoding for carbohydrate active enzymes, can impede their differentiation if such genes are targeted in complex microbiomes such as environmental samples. Most importantly, D1–D2 region has been lately more accepted as the reference DNA region for anaerobic gut fungi and thus provides curated reference data that can be used for classification and phylogenetic inference [1,13,14].

So far, no specific PCR primer pair enables simultaneous *Neocallimastigomycetes* quantification by qPCR, amplification from species-rich environmental samples, and sequencing by short read high-throughput techniques with the same amplicon. Such a primer pair must be specific and should amplify a DNA segment that is short enough to allow HTS and qPCR while also providing adequate phylogenetic resolution.

In the current study, a newly developed primer pair targeting the *Neocallimastigomycetes* rDNA Large Subunit D2 region (D2 LSU) was developed and evaluated. Due to its specificity, adequate length, and sufficient degree of phylogenetic resolution, it can be applied to environmental samples with highly diverse microbial communities allowing quantitative analyses and conjointly assessing the *Neocallimastigomycetes* community composition mostly to the species level using HTS methods. As a part of the evaluation, we aimed at detecting novel *Neocallimastigomycetes* clades in the examined environmental samples.

## 2. Materials and Methods

### 2.1. Primer Design and Validation

Approximately 2600 sequences from the D1–D2 region of LSU were compiled and aligned using different versions (v.6, v.7, v.MX) of the software package MEGA [20]. The sequences were retrieved from open databases such as the National Center for Biotechnology Information (NCBI, https://www.ncbi.nlm.nih.gov/, accessed from October 2019 to February 2022), the Joint Genome Institute’s Genome Portal (JGI, https://jgi.doe.gov/, accessed from October 2019 to February 2022), and recently described genera published by Hanafy et al. [15]. Further sequences were obtained from AGF isolated from different animals at the Bavarian State Research Center for Agriculture (LfL), the Department of Microbiology and Molecular Genetics of the Oklahoma State University (OSU), and other sources (Table 1). To design specific primers, the most closely related sequences were identified by nucleotide BLAST [21] against the database nucleotide collection (nr/nt) and additionally added to the alignment. The outgroup comprised various sequences of *Chytridiomycota*, e.g., DQ536493 *Chytriomyces* sp. WB235A. 

From this comprehensive D1-D2 alignment, primers with a similar melting temperature (T_m_) were designed in silico using Primrose 2.17 [22], and the proposed positions were carefully checked manually for suitability with the premise that the 3′ primer ends were signature specific for *Neocallimastigomycetes*. In total, nine potential primer binding sites were evaluated in silico within the ribosomal LSU (see Appendix A). Eventually, the primer pair AGF-LSU-EnvS was selected (see results section) and evaluated for their performance. TestPrime (https://www.arb-silva.de/search/testprime/ accessed on 6 October 2020) was used to screen the number of mismatches (consecutive from zero to four mismatches) for the primer combination until the first outgroup was detected in the SILVA LSU reference database (LSU r138.1, [23]).

### 2.2. Cultivation of Pure Cultures

Representatives from sixteen out of the twenty described *Neocallimastigomycetes* genera were obtained from different sources to cover as many isolates as possible and test the primers in vitro. Table 1 shows the examined isolates/DNA extracts and their providers. 

The LfL culture collection, which includes the isolates from DoLS, UIBK, and IAPG (Table 1), is kept routinely in anaerobic basal medium prepared according to Callaghan et al. [11]. Dissolved xylan solution (0.2% *w*/*v*), cellobiose (0.2% *w*/*v*), and wheat straw milled to 1–2 mm (0.5% *w*/*v*) were added as carbon sources. Fifty mL of the CO_2_ gassed basal medium was aliquoted in 120 mL serum bottles and hermetically closed with butyl rubber stoppers and aluminum caps to keep an anoxic atmosphere inside. An antibiotic solution of penicillin sodium salt, ampicillin sodium salt, and streptomycin sulfate (0.5 mg/mL each in the final medium) was injected into each serum bottle prior to inoculation to suppress bacterial growth.

The isolates were grown for three rounds of subculturing on medium without wheat straw, in which the clarified rumen fluid (CRF) was replaced per liter media with 10 mL trace metal solution [26] and 10 mL vitamin solution [27]. After incubation of 3 to 5 days from the last round of subculturing, the biomass was harvested by centrifugation at 7197× *g* for 5 min. The supernatant was discarded, and the biomass pellet was immediately used for DNA extraction. In addition, DNA was extracted from the aquatic fungus *Chytridiomycetes* sp. isolate AUS_17 (grown on PmTG agar) to serve as a negative control.

### 2.3. Environmental Samples

The PCR-based protocols were tested on environmental samples consisted of three groups: animal feces where AGF have been shown to be present, animal feces from carnivores and omnivores where AGF were not expected to be present, and primarily animal-unassociated and highly diverse habitats such as soil and biogas plant digestates (Table 2). 

### 2.4. DNA Extraction

Cultivation and DNA extraction from pure cultures owned by KIT, OSU, and ZHAW were performed by the owners according to their lab protocols referenced in Table 1 [1,24,25], and the eluted DNA was sent to LfL. Additionally, DNA from the three types of samples (pure cultures, feces, and environmental samples as soil and digestates) were extracted at LfL (Table 1). Pure cultures were harvested by centrifuging at 7197× *g* for 2 min and repeatedly centrifuged with sterile 0.85% KCl solution to wash until the supernatant was clear. Approximately 90 mg of each washed pure culture biomass pellet was homogenized in Lysis Matrix E tubes (MP Biomedicals, Irvine, CA, USA) with a FastPrep-24 homogenizer (MP Biomedicals) for 20 s and a speed of 4.0 m/s. DNA was extracted according to the DNeasy^®^ Plant Pro kit (Qiagen, Hilden, Germany) protocol. The same protocol was used for the feces samples, but these were homogenized with a vortex adapter (Qiagen cat. nr. 13000-V1-24) at maximum speed for 10 min.

Soil and biogas plant digestates were homogenized in Lysis Matrix E tubes (MP Biomedicals) by a FastPrep-24 homogenizer (MP Biomedicals) for 20 s and a speed of 4.0 m/s. In the case of the soil sample, 400 mg were washed with 0.85% KCl solution, and the DNA was extracted with phenol/chloroform/isoamyl alcohol according to Griffiths et al. [28]. Forty µg of the biogas plant digestate samples were used for DNA extraction following the GeneMATRIX Universal DNA/RNA/Protein Purification Kit (EurX, Gdańsk, Poland) protocol. All extracted DNA was stored at −20 °C until further use for downstream applications such as cloning, sequencing, conventional PCR, and qPCR.

The different DNA extraction protocols were selected according to the best practices and results routinely obtained for each sample type at LfL. The fresh weights of biomass/material used for each DNA extraction were taken to normalize quantification results. If the same extraction protocol was used, comparing samples was possible.

### 2.5. Polymerase Chain Reaction (PCR)

All PCR reactions were performed with the primer pair AGF-LSU-EnvS and contained 2.5 µL 10 × PCR buffer, 6 mM MgCl_2_ (50mM), 200 µM dNTPs (10 mM each), 250 nM of each primer, and 0.75 U Platinum^TM^ Taq DNA Polymerase (Life Technologies, Carlsbad, CA, USA), and 1 µL DNA template (concentrations between 0.6 and 17.5 ng/µL) in a total reaction volume of 25 µL. In no template controls (NTC), the DNA template was replaced by nuclease-free water. The PCR program consisted of 3 min initial denaturation/activation at 95 °C, 35 cycles of 20 s denaturation at 95 °C, 30 s annealing at 62 °C, and 60 s elongation at 72 °C. A final extension of 10 min at 72 °C concluded the PCR run with a hold at 4 °C thereafter. The PCR runs were performed on a TProfessional Thermocycler (Biometra, Jena, Germany) or a Flexcycler (Analytik Jena, Germany). Agarose gel electrophoresis (1% *w*/*v* agarose in TAE buffer, 3 µL PCR product, 60 min, voltage of 100 V, 2 µL SERVA DNA Stain G dye) was used to visualize the PCR products.

### 2.6. Quantitative Real-Time PCR (qPCR)

DNA extracts were quantified by qPCR using the AGF-LSU-EnvS primer pair and identical PCR reagents as above, replacing 1 µL water with EvaGreen Dye (4 µM, Jena Bioscience, Jena, Germany). All qPCR runs were performed in a two-step program on an AriaMx Real-Time PCR (qPCR) Instrument (Agilent, Santa Clara, CA, USA). The assay consisted of an initial denaturation/activation step at 95 °C for 3 min followed by 40 cycles of 15 s denaturation at 95 °C, 1 min annealing/extension at 62 °C, and 10 s denaturation at 81 °C (data acquisition was taken after this step to exclude potential unspecific products with a lower T_m_). The specificity of the amplification was inspected by a melting curve analysis (1 min denaturation at 95 °C, 30 s annealing starting at 55 °C with 0.5 °C ramping per 5 s, and 30 s at 95 °C for final denaturation) that typically shows a specific T_m_ peak around 84.5 °C. All samples, NTCs, and standard reactions were run and analyzed in technical triplicates. Tenfold dilutions of each sample were additionally analyzed. The cycle of quantification (Cq) value slopes of about −3.3 confirmed the absence of any PCR inhibiting effect for all tested samples [29].

A DNA dilution series of a pure culture of *Neocallimastix cameroonii* isolate YoDo11 (Table 1) was used as quantification standard in each assay. The standard was 10-fold diluted until 10^−5^, where no positive signal on the triplicates was detected anymore, and calibrated in an MPN approach according to the method of Lebuhn et al. [29] and Munk et al. [30]. The standard equation Y = −3.278 × LOG (X) + 37.09 (Y = Cq value; X = initial quantity of copies), with a 0.998 coefficient of determination (R^2^) and a 101.86% efficiency was obtained from the standard curve dilution series, indicating efficient amplification and reliable quantification [29]. The Cq values of the samples and NTCs were quantified with the standard curve formula specified above. The quantification values were corrected by subtracting the corresponding NTC copy numbers (Cq values equivalent to 5.4 to 22.6 copies), and the sample yields. Standard errors were calculated for all technical replicates. In order to correct the results due to losses during the extraction procedure, the extraction efficiency per DNA extraction method was considered (GeneMATRIX Universal DNA/RNA/Protein Purification Kit (Roboklon, Berlin, Germany): 90%; DNeasy^®^ Plant Pro kit: 75%). The extraction efficiency of the DNeasy^®^ Plant Pro kit (Qiagen) was assessed by spiking a known DNA amount of one *Neocallimastigomycetes* isolate in an autoclaved soil sample. The spiked sample was run through the manufacturer’s protocol. The loss factor was calculated, and the amount of elution buffer was considered to correct the quantifications. The efficiency of the Roboklon kit was determined by comparison of results for given samples using the MP biomedicals FastDNA™ SPIN Kit for Soil [29] in parallel. The target copy number was determined per gram fresh weight or mL of the sample.

### 2.7. Amplicon Sequencing

The AGF-LSU-EnvS amplification of environmental fecal samples from AGF hosts was evaluated by sequencing of the PCR products and analyzing them phylogenetically. Clone libraries and HTS by Illumina were used to determine the composition of the mixed AGF communities and amplicon sequencing variants (ASV). Clone libraries of amplicons from animal feces were generated via cloning with the TOPO-TA Cloning^®^ Kit for Sequencing pCR^®^ 4-TOPO^®^ Vector (Invitrogen, Waltham, MA, USA) following the manufacturer’s protocol. Clones with inserts showing the expected length in agarose gel electrophoresis were sequenced unidirectionally (Eurofins Genomics via Sanger method). The resulting sequences were manually inspected against the chromatograms and corrected if necessary. 

In addition, amplicons for Illumina short-read sequencing were generated with the AGF-LSU-EnvS primers elongated at the 5′ end with respective standard Illumina adapters: forward 5′-TCGTCGGCAGCGTCAGATGTGTATAAGAGACAG-3′ (33 bp) and reverse 5′-GTCTCGTGGGCTCGGAGATGTGTATAAGAGACAG-3′ (34 bp). The sequencing was performed on an Illumina MiSeq device using the v3 chemistry (2 × 300 bp) at the ZIEL Core Facilities (Institute for Food & Health, Technical University of Munich, Germany) following the protocol described by Reitmeier et al. [31]. The resulting paired-end sequences were approx. 420 bp in length including primers and adapters. 

### 2.8. Mock Communities to Test the Data Processing Pipeline

The reliability of the Illumina sequence analysis pipeline was evaluated by testing artificial mock communities. DNA from 18 pure cultures was amplified with the AGF-LSU-EnvS primer pair as described above. The amplicon concentrations were measured by a Qubit 4.0 fluorometer (Thermo Fisher Scientific, Waltham, MA, USA) with the Qubit dsDNA Broad Range Assay-Kit (Thermo Fisher Scientific). The PCR products were subsequently pooled in defined relative proportions to simulate different AGF communities: Mock community 1 (10% *Caecomyces communis* var. *churrovis* and 5.3% per each of the other 17 taxa) depicted several AGF species at similar abundance (high rich evenness and high diversity), and Mock community 2 (low diversity and low evenness) contained only two species *Khoyollomyces ramosus* as dominant (90%) and *Agriosomyces longus* (10%) in low abundance.

### 2.9. Sequencing Data Processing

The Sanger sequences were screened for chimeric sequences in two steps: first, they were checked by Uchime in Mothur [32] in de novo mode (xn = 5; minchunk = 32). In a second step, after aligning the sequences with the references (see section “Primer design and validation”), they were ordered according to their topology in a neighbor joining guide tree, where further potential chimeras typically are found in a position between significant clades, and their parents can easily be identified visually.

A QIIME2 [33] pipeline was used to analyze the produced Illumina sequences. All raw amplicon sequencing datasets were imported into the QIIME2 platform as a zipped data format called artifact. Once imported, the artifact files can be processed by each QIIME2 module. In the first step of the pipeline, the raw sequencing data were demultiplexed into each individual sample based on its identification barcode sequence. The demultiplexed datasets were denoised using the DADA2 [31] implementation in the QIIME2 pipeline (with input parameter: --p-trim-left-f 15 --p-trim-left-r 15 --p-trunc-len-f 240 --p-trunc-len-r 160 --p-min-fold-parent-over-abundance 4). In the denoise step, pair-end amplicon sequences are filtered, trimmed, and merged into unique amplicon sequence variants (ASVs). In the denoising process, DADA2 also tries to remove chimera sequences using a de novo-based approach. The de novo-based chimera detection utilizes high abundance ASVs as reference sequences to probe for hybrid sequences that were potentially generated during the PCR step. On top of DADA2, we also used VSEARCH [32] with a stricter parameter (--p-mindiv 0.2 --p-minh 0.05) to further remove chimera sequences. The rest of the sequences were manually checked for chimeras, as the remaining ASVs were in a manageable amount.

Cluster analysis was done using the software packages R and RStudio [34]. The raw ASV data were normalized to the smallest number of total reads in one sample (lama, 13,052 reads) using rrarefy in the R package vegan [35]. A heatmap was generated with the function heatmap.2 in the gplots package [36] using standard parameters which includes computation of the distance (vegdist method: bray) and cluster analysis with hclust (method: average).

### 2.10. Phylogenetic Analysis

From the compiled 2600 sequences from the D1–D2 region, high quality non-chimeric and non-redundant sequences were selected to create reference files. All generated D2 LSU sequences (Sanger and Illumina sequencing) were aligned to these references by ClustalW [37] using MEGA 6 [20] and manual refinement. Neighbor-joining trees of the D1–D2 region and the D2 domain were calculated applying the Tamura 3-parameter model, pairwise deletion of gaps, and 100 bootstrap iterations as implemented in the software package MEGA 6 [20]. *Chytriomyces* sp. WB235A (accession number DQ536493; phylum *Chytridiomycota*) was used as outgroup. In order to describe the community structure of *Neocallimastigomycetes* in the environmental samples, the taxonomic affiliation of the sequences was inferred. A mean sequence identity threshold of about 98.5% was applied to delineate species and about 95% to discern genera [38]. These criteria were used to annotate the taxonomy of the environmental sequences in the reference framework. Optimum resolution was obtained by blasting questionable sequences against the longer and more informative reference alignment composed of sequences spanning the LSU D1 and D2 domains.

## 3. Results and Discussion

### 3.1. In Silico Primer Search

The fundamental goal of this study was to identify a primer set that is specific for *Neocallimastigomycetes* and amplifies a region that can be used as a barcode in environmental samples. Therefore, the divergent regions of the LSU were screened for suitable primer sites. In total nine sites were evaluated against an alignment with 2600 sequences of the D1–D2 LSU. Besides testing the primers, the analysis of the 2600 sequences of the D1–D2 LSU alignment revealed, as demonstrated by manual inspection of the original Sanger sequence chromatograms, that the two *Capellomyces elongatus* GFKJA1916 accession numbers MT085701 and MK775304 sequences had a compression at the highly conserved envisaged reverse primer signature position where a double instead of a single T should have been edited at position 1390 of MT085701 and 740 of MK775304. Such sequencing artifacts are not uncommon at sequence ends [39].

Among the 720 aligned sites of the D1–D2 LSU region, 373 (51.8%) were parsimony-informative. From the 348 sites constituting the D2 region, 252 (72.4%) were parsimony-informative, indicating a high-resolution potential of the D2 region. The D2 region is located at the beginning of the LSU after the D1 region of the *rrn* operon. For a detailed schematic diagram showing the D2 region position, see Edwards et al. [12], who indicates *Orpinomyces* sp. OUS1 (AJ864475) operon arrangement.

The primer pair AGF-LSU-EnvS that targets the D2 region of the LSU showed the best specificity for all currently known *Neocallimastigomycetes*. The best forward primer was obtained from the Primose 2.17 search (allowing 3 mismatches per primer), and the best reverse primer was identified to be a modification of the primer GGNL1 [18]. The next detected outgroup by TestPrime against the SILVA reference database was *Rhizophydium brooksianum* (*Chytridiomycota*) with four mismatches for the forward and one mismatch for the reverse primer. The characteristics of the primer pair AGF-LSU-EnvS are described in Table 3.

There are several advantages of applying the proposed AGF-LSU-EnvS primer pair compared to previously published methods:(i)The gap in coverage of AGF is closed compared to earlier reported primer sets, e.g., AGF-LSU reverse [14] did not detect *Aestipascuomyces*, *Agriosomyces*, *Buwchfawromyces,* and *Joblinomyces;* GGNL1 forward [18] did not match with newly described AL3 and AL8 clades [15], and GGNL4 reverse did not match with *Anaeromyces contortus*.(ii)In contrast to earlier methods where detection, quantification, and classification needed several steps with different primers, here only one genetic marker is used, avoiding bias by different primer selectivities.(iii)The alienability of the D2 LSU amplicon presents advantages in comparison with any ITS amplicon and its intra-strain heterogeneity is much lower in the LSU than in the ITS regions where the differences frequently exceed species limits.

### 3.2. Detection of AGF in Cultures and Environmental Samples

PCR reactions with 20 isolates of 18 different AGF species (Table 1) showed that the primer set AGF-LSU-EnvS successfully amplified the target region of all the pure cultures tested (Figure 1). The weak amplification of *Anaeromyces contortus* G3C, *Capellomyces foraminis* BGB-12, *Caecomyces communis* var. *churrovis* YoDo26, and *Khoyollomyces ramosus* ZC-41 was due to the low template DNA concentrations (<2 ng/µL). To confirm that the template concentration was the issue and not the primer binding affinity, other isolates of *Caecomyces communis* var. *churrovis* (ViSuPo1A) and *Khoyollomyces ramosus* (H1WF70RF2), with higher DNA concentration (>6 ng/µL), were included in the same PCR run. As expected, the bands (Figure 1) were more intense, indicating adequate primer affinity. *Capellomyces foraminis* and *Anaeromyces contortus* could not be tested again because enough DNA extract or alternative isolates were not available.

The DNA extracts of environmental samples from feces of different herbivores also produced specific amplicons of the correct length (Figure 2). Even when each fecal sample was composed by a different AGF community (see “Community Analysis” below), all bands were similar.

As expected, the pig manure, dog feces, and soil samples did not show amplification (Figure 3). DNA extract from the aquatic fungus *Chytridiomycetes* sp. isolate AUS_17 (order most closely related to *Neocallimastigomycetes*) was also negative, supporting the high specificity of the primer pair in vitro (Figure 3). DNA extracts obtained from cow manure and *Feramyces austinii* isolate DF1 were used as positive controls in the same PCR run.

The digester sludges of biogas plants PB 8 (fed with energy crops but no cattle manure) and PB 25 (fed with 36% cow manure, a potential habitat for AGF) did not show amplification in the conventional PCR (Figure 3). As the PB 25 was a sample related to cow manure, a qPCR assay was run for this sample to detect if low concentrations of AFG are present in the sample. As expected, the quantification analysis showed that biogas plant PB 25 sample presented a positive and specific AGF signal. This same sample was screened by Dollhofer et al. [7], also identifying AGF amplicons in it. Dollhofer et al. [7], observed that biogas plants fed with ≥30% cow manure presented qualitative and quantitative results of AGF. The lack of a band on the agarose gel from the conventional PCR would be a result of a low *Neocallimastigomycetes* DNA concentration (see “Quantitative Real-Time PCR (qPCR)” below), thus, the use of qPCR is recommended to generate more detailed results if uncertainties after conventional PCR analysis arise. 

The in silico and in vitro validations of the proposed primers confirmed specific detection of the D2 LSU region of *Neocallimastigomycetes* isolates and corresponding environmental samples. In silico analyses showed that in the case of *Capellomyces foraminis*, both primers matched 100% of the sequences deposited in NCBI, whereas most probably due to sequence editing artifacts caused by a double thymine (TT) compression at the sequence ends, a theoretical mismatch appeared at position 12 (in the middle) of the reverse primer for the two *Capellomyces elongatus* GFKJA1916 sequences (MT085701, MK775304) and for three out of 34 *Anaeromyces contortus* sequences (*Anaeromyces* sp. NHY-2018, MF121936, MF121942, and MF121941). Besides such assumed sequence editing errors, both primer sequences are 100% conserved among the known *Neocallimastigomycetes* and specific for this class, considering the current knowledge. Further analyses of the complete *rrn* operons are ongoing and may confirm our assumptions.

### 3.3. Quantitative Real-Time PCR

The *Neocallimastigomycetes* LSU DNA copy numbers were quantified for four samples applying the primer pair AGF-LSU-EnvS. Genomic DNA were extracted from feces of cow (as model host with expected positive signal), elephant, lama (randomly chosen samples), and mara (less explored habitat). These four samples produced specific qPCR signals in the range of the standard curve of DNA from a *Neocallimastix cameroonii* isolate (Figure 4A), Cq (cycle of quantification) values below those of the no template control (NTC, mean Cq value of 33.78) reactions, and a melting curve peak between 84.5 °C and 85.0 °C (Figure 4A,B) indicated specificity of the reactions. The melting curve analyses revealed a difference between the mara feces DNA and the other tested samples. The peak generated by mara feces DNA had a T_m_ of 84.5 °C (peak A in Figure 4B), whereas the curves of the other samples, including the *N. cameroonii* DNA, peaked at 85.0 °C (peak B in Figure 4B). This shift reflects that the amplified sequences from mara feces have a different nucleic acid content (CG%), therefore suggesting that the AGF community may differ from *N. cameroonii* (used as control), cow, elephant, and lama feces samples tested in the same qPCR run.

The copy number of *Neocallimastigomycetes* LSU DNA in environmental samples varied from 8.3 × 10^8^ copies per gram fresh weight (FW) in lama feces, to 1.5 × 10^11^ copies/g FW in mara feces. For the AGF cultures, between 1.4 × 10^11^ copies/g FW for *Feramyces austinii* DF1 and 3.0 × 10^12^ copies/g FW for *Buwchfawromyces eastonii* GE09 were obtained. The average of the LSU copy number of the four environmental (fecal) samples was at 5.6 × 10^10^ copies/g FW (SD = 1.5 × 10^10^), meanwhile the pure cultures averaged at 7.8 × 10^11^ copies/g FW (SD = 1.23 × 10^11^), which is about ten times more than the average of the environmental fecal samples.

The samples where no AGF were indicated by the conventional PCR (Figure 3) were further quantified by qPCR to corroborate that there are no AGF even in a low concentration and to detect potential unspecific signals. The DNA samples from pig manure, dog feces, digester sludge from biogas plant 8 (PB 8), soil, and the *Chytridiomycota* isolate (Chytrid) presented low non-specific signals (T_m_ of 77 °C to 79.5 °C), as well as Cq mean values close to or even higher than the NTC (NTC Cq = 33.78, soil Cq = 35.9, pig manure Cq = 32.9, dog feces Cq = 34.7, PB 8 Cq = 33.8, and Chytrid Cq = 35.1). This indicated that AGF were absent in these samples, and that Cq values below 30 can be safely quantified as AGF. DNA from cow manure and *N. cameroonii* used as positive controls in the same qPCR run presented specific signals (T_m_ 84 °C to 85 °C as seen in Figure 4B, and low Cq values between 14 and 25). Although no band from conventional PCR was visible on the agarose gel for digester sludge of biogas plant 25 (PB 25) (Figure 3), the corresponding qPCR amplicon was specific (84 °C) and slightly positive (Cq < NTC), with 2.9 × 10^6^ copies/mL. This biogas plant was fed with approximately 36% cow manure, suggesting that traces of *Neocallimastigomycetes* DNA were present in the digester sludge as a relic of the cow manure feedstock, and that the qPCR analysis was more sensitive than the conventional PCR analysis.

The D2 LSU results obtained in this study were compared with those from Dollhofer et al. [7,14] who used the SSU rDNA region. Both regions are part of the ribosomal operon and should hence have the same copy number. Dollhofer et al. [7,14] reported SSU values for cattle rumen fluid of 1.69 × 10^10^ ± 3.88 × 10^9^ copies/mL which was similar to the D2 LSU copy numbers in the fecal samples (5.6 × 10^10^ copies/g) and up to 1.65 × 10^9^ SSU copies/g in digestates from different biogas plants. The biogas plant PB 25 sample (2.96 × 10^6^ D2 LSU copies/g FW) had also been analyzed by Dollhofer et al. [14] and showed 1.78 × 10^8^ SSU gene copies per ml fermenter sludge in their study. The considerably lower value can be explained by the long storage time of this sample, the decay of the organisms, and the disintegration of the nucleic acids.

Pre-screening samples from other potential AGF habitats and interesting AGF communities by specific (q)PCR-based analyses can be very useful for saving time and costs. Original community studies and identification of species typically require isolation and cultivation of AGF. These are still demanding tasks, particularly if several potential habitats are being explored simultaneously. In this case, the isolation medium typically uses rumen fluid or simulates a rumen environment, but the respective conditions can be unsuitable to enhance the whole diversity from non-rumen habitats for example [25]. Thus, (q)PCR assays can promptly indicate if it is worth starting isolation/cultivation procedures, and if so, more time can be spent on optimizing isolation depending on the type of sample to analyze, for example the use of rumen fluid or not in the isolation medium from environments outside the rumen.

### 3.4. Evaluation of the AGF-LSU-EnvS Processing Pipeline Using Mock Communities

The bioinformatics pipeline generated high quality, chimera-free ASVs from the two different mock communities with various levels of diversity and evenness. The isolates that were artificially assembled in these mock communities were detected at similar ratios after running through the DADA2 pipeline with QIIME2, indicating that no major errors occurred (Table 4). However, some differences and minor issues should be mentioned. In Mock community 1, *Aestipascuomyces dupliciliberans, Anaeromyces mucronatus,* and *Capellomyces foraminis* were detected in a slightly lower ratio than expected (approx. 3.5% with a standard deviation of 0.2%), and *Paucimyces polynucleatus* was detected in a slightly higher ratio (6.9%). The other isolates were detected correctly with about 5.3% ± 0.5%, and *Caecomyces communis* var. *churrovis* with an expected ratio of approximately 10.5%.

Mock community 2, where *Khoyollomyces ramosus* and *Agriosomyces longus* were pooled in a 9:1 ratio, presented also very accurate results (Table 4). The composition of a remaining 0.7% of the sequences, however, was unexpected, as no *Piromyces* sp. (0.6%) or *Anaeromyces contortus* (0.1%) were added to the pool, thus pipetting errors and contamination may have occurred to a negligible extent. Therefore, it is suggested that very low amounts of reads (<0.7%) should be analyzed individually or in practical cases a threshold can be established. In summary, the proposed protocol through the pipeline appears to be adequate to detect, quantify and classify *Neocallimastigomycetes* targeting their D2-LSU amplicon in complex communities. The variable number of ASVs should be noted though, as it implies that the community may appear richer than it is, but these ASVs belong to the isolates purposefully included in each mock community. Additionally, future studies will yet have to show, whether this ASV per strain diversity is constant for a species or not. The variable number of one to four ASV per strain will be of significance for future environmental screenings, which will require an additional clustering step to avoid inflating diversity analysis and to allow working on a taxa level. Here, an additional clustering step to OTUs using a hard or a dynamic cutoff may be a good solution.

### 3.5. Community Analysis

The community composition of AGF sequences amplified by the primer set AGF-LSU-EnvS was analyzed using Sanger sequencing of clone libraries and HTS by amplicon sequence variants (ASVs) from Illumina sequencing (Table 2). DNA extracts from feces of cow (as model host), elephant and lama (randomly chosen samples), and mara (unexplored host) were examined by cloning/Sanger sequencing. In total, 169 clones were produced from these four feces samples. These four samples and six other animal feces samples were analyzed by HTS (Table 2), resulting in a total of 258,643 reads sorted in 129 unique Illumina ASVs. Illumina sequencing produced a detailed representation of ASVs per species, not just by describing the species present in each sample but also by detecting different ASVs per species: *Cyllamyces* spp. comprised the highest amount of ASVs (*n* = 29), followed by *Orpinomyces joyonii, Neocallimastix frontalis* (*n* = 11), and *Anaeromyces mucronatus* (*n* = 10). 

To compare the performance of commonly sequencing tools used, feces from four animals (cow, elephant, lama, and mara) were selected for cloning/Sanger sequencing to be compared to their Illumina sequencing results. The distribution patterns of the clone library showed high similarities to the distribution of the Illumina ASVs (Appendix A). However, *Paucimyces* spp., *Khoyollomyces ramosus*, *Caecomyces communis* var. *churrovis,* and some potential new genera/species from the lama feces were not detected by Sanger sequencing. In contrast, few sequences were detected by Sanger but not in the ASVs, for example the clones from the genera/species *Piromyces* sp. 6 (ASV0033) from cow and lama fecal DNA (Appendix A). This indicates that cloning/Sanger sequencing can also be used in case next generation sequencing technologies are not available. Nevertheless, the later technology is recommended to get a better view of the AGF community in environmental samples.

The amplification of the LSU D2 region in the DNA extracted from feces of ten different herbivores showed that AGF were present in all analyzed samples (Figure 2). Their Illumina sequencing results gave a very detailed picture of the AGF community composition in such fecal samples and revealed different communities for the individual host animals (Figure 5). Some ASVs were found in several animals, demonstrating that one species can inhabit different animals. *Caecomyces communis* var. *churrovis* and *Neocallimastix frontalis* were found in almost all sampled hosts, while *Cyllamyces aberensis*, *Feramyces austinii*, and *Khoyollomyces ramosus* were only found in a particular host: alpine ibex, bison, giraffe, and horse, respectively.

The case of the genus *Piromyces* is of particular interest because different species were found in the individual animals, suggesting that distinct *Piromyces* species may be specific for a particular host: *Piromyces communis* was found in Alpine ibex (716 reads), *Piromyces* sp. 4 in cow (1191 reads), and *Piromyces* sp. 6 in elephant (3481 reads). Cand. *Piromyces potentiae* was composed by seven ASVs, six of which were only present individually in separate particular hosts (alpaca, lama, bison, cow with 2 ASVs, and elephant), and only one ASV (ASV0036-Cand. *Piromyces potentiae*) was found in three different hosts (lama, alpaca, and kangaroo) (Appendix A).

The AGF community in the 10 different animals can be divided into two groups. In the first group, comprising the fecal samples with a diverse AGF community composed of more than two species and all species in a relative abundance lower than 60% (cow, lama, elephant, alpine ibex, alpaca, and bison). The second group comprises fecal samples with a simple AGF community dominated by only one species in a relative abundance >95% ASV reads per species (mara, kangaroo, horse, and giraffe; Figure 5). Both groups are integrated by animals with different digestive tracts, this indicates that the physiological characteristics of the digestive tracts from the animal hosts dos not influence the AGF community composition. 

Potentially new *Neocallimastigomycetes* genera/species were detected in the fecal samples from kangaroo and mara (*Neocallimastigaceae* clade YL5 and *Neocallimastigaceae* clade YL2, respectively; Figure 5). The potentially new *Neocallimastigaceae* clade YL2 (mara) was previously detected by the qPCR results by a shifted melting curve of the mara feces sample to a T_m_ of 84.5 °C (Figure 4B). This suggested already that the sequences derived from mara feces differ in GC content from the other samples (T_m_ = 85.0 °C) that were run in the same experiment (feces of lama, elephant, and cow, and *Neocallimastix cameroonii* as pure culture). This difference in the AGF community composition is corroborated by the results of the Illumina amplicon sequencing (Figure 5) and the evaluation of the clone libraries, where the mara amplicons formed a well-distinguished new clade, branching off at the genus level (“*Neocallimastigaceae* clade YL2”, Figure 6 and Figure 7). 

The presence of AGF in mara feces was previously reported by Teunissen et al. [41] using isolation/cultivation methods. Their short description mentions that the AGF found were similar to *Piromyces*. As no molecular tools were used to determine its classification, and morphological determination can be misleading for many AGF, it is not conclusive if *Piromyces* and/or other genera or species were present in their mara feces. *Piromyces* has been found also in the kangaroo digestive tract [42], but the case is similar to AGF in mara, where not much information can be found and the available records were published in the decade of 1990. According to our findings and the few information published regarding explored AGF in kangaroo and mara, they are good examples of environments which deserve more detailed investigation.

In comparison with kangaroo and mara, cow and horse digestive tracts are much more studied, and several genera living in them have been described, not just morphologically, but also with molecular tools using ITS and LSU [43,44]. *Neocallimastix*, *Orpinomyces*, and *Anaeromyces*, as well as uncultured *Neocallimastigomycetes* have been previously found along the horse digestive tract [43]. In this study, and according to our comprehensive LSU D1–D2 reference file, these uncultured sequences can be affiliated as *Piromyces finnis* and *Khoyollomyces ramosus* (MH125212–MH125228 from [43] in Appendix A). In our findings *K. ramosus* was found as the main species in horse feces. Advanced knowledge of novel diversity can pave the way to isolation and cultivation of undiscovered AGF species as well as to enrich reference files to improve taxonomic classification.

Our results give hints to a correlation between a low number of *Neocallimastigomycetes* species present in monogastric animals (horse, mara) and animals with fewer compartments along the digestive tract (kangaroo), than in ruminants which have several compartments (Table 2). This applied to eight out of ten host species examined in this study. However, giraffe, with four compartments, and the monogastric elephant, are exceptions to this pattern. Giraffe has been previously sampled searching for *Neocalimastigomycetes*, were *Neocallimastix*, *Orpinomyces*, and *Piromyces* were found in giraffes digestive tracts [45]. 

Another interesting observation is that the communities composed of only one species were not found in other hosts. For example, the “single-species” *Neocallimastigomycetes* communities in mara, horse, kangaroo, and giraffe feces are different between each other and seem to be specifically found in the respective host. Conversely, none of the hosts with numerous *Neocallimastigomycetes* species contained in considerable amounts any of the unique species present in mara, horse, kangaroo, and giraffe. The protocol using the AGF-LSU-EnvS primer pair would make it easy to examine more animals to test these theories and to examine the effects of domestication in different animal groups.

One additional benefit of having the AGF-LSU-EnvS primer pair as qPCR assay and as amplicon for sequencing-based studies is that it will be possible to scale the compositional sequencing data by molecular abundance. While care must be taken when the ribosomal copy number is too divergent, it can be used to differentiate between pseudochanges and real changes in the relative abundances of AGF taxa in environmental samples [46].

### 3.6. Phylogenetic Analysis

Any meaningful phylogenetic analysis is dependent on the resolution capability of the available sequences and on the completeness and reliability of the reference data against which a query is directed. The original alignment to test the primers comprised 2600 aligned chimera-free high quality D1–D2 LSU sequences of the known genera, species and proposed but unassigned clades. From this alignment, 854 non-chimeric non-redundant sequences were selected to compose the D1–D2 LSU reference file used for phylogenetic analysis. Figure 6 presents the respective D1–D2 LSU phylogenetic neighbor joining tree. All clades are compressed to improve the visualization, but the complete extended phylogenetic tree is added as Appendix A. This D1–D2 LSU phylogenetic tree clearly shows several novel clades. These new clades can be divided into two groups: One group comprised new clades formed by the whole D1–D2 region sequences resulting from the present work namely: *Orpinomyces* sp. 4 (ON758335), *Neocallimastigaceae* clade NC3 (ON619900, ON619903, and ON619902), *Neocallimastigaceae* clade YL2 (Mara, ON650566 to ON650603), and *Neocallimastigaceae* clade YL1 (ON758305 to ON758307).

The second group of new clades is made up of sequences published by Hanafy et al. [15]. These sequences depict the following new clades also shown in Figure 6: *Orpinomyces* sp. 3, *Neocallimastigaceae* clade RH2, *Caecomyces* IA sp. 2, *Neocallimastigaceae* clade AL8, *Joblinomyces* sp. 2, and even a deep branching (potentially new family) in the *Neocallimastigomycetes*, called clade AL3 (Appendix A). Interestingly, the *Piromyces* clade showed a wide subdivision in subclades. The D1–D2 tree shows 8 subdivisions inside the genus (Figure 6), with 5 potentially new clades marked as *Piromyces* sp. followed by consecutive numbers.

The D1–D2 LSU reference file and the respective phylogenetic tree were used to affiliate the amplicons generated in this work. The shorter D2 LSU sequences (Sanger and Illumina) generated with the AGF-LSU-EnvS primer set were classified phylogenetically by blasting against the D1–D2 reference file and are shown in a second neighbor-joining tree (Figure 7). The new genera and species were based on the suggestions and rank assignment criteria published recently [38]. These indicated the separated topology and considerable phylogenetic distance of several sequences and clades from the animal feces (Figure 7). The proposed new genera (<95% identity with nearest known neighbors) are shown in Figure 6, and the potential new species (<98.5% identity with nearest known neighbors) are listed as genera/species (sp., with numbering) in Figure 6 and Figure 7. Besides these novel discoveries, known and well-described genera such as the extensive *Piromyces* clade deserves attention, as the results of the phylogenetic analyses (Figure 6 and Figure 7) suggest the presence of several new genera/species.

The new clades appearing in the D2 phylogenetic tree (Figure 7) as compared with the D1–D2 tree (Figure 6) are: *Paucimyces* sp. 2, the *Neocallimastigaceae* clades YL1 to YL5, *Caecomyces* II (not significant), some *Neocallimastigaceae* inc. sed. (related to *Neocallimastigaceae* clade YL1), and some minor branches. The extended *Neocallimastigaceae* clade YL1 may even be grouped into 4 or more species (Figure 7).

The comparison of Figure 6 with Figure 7 shows that due to the lower number of informative sites (shorter amplicon), the exclusive use of the D2 region cannot equal the higher resolution of the phylogenetic tree generated with the longer amplicon of the LSU D1–D2 region. Apart from the notoriously problematic genera *Orpinomyces* and *Caecomyces/Cyllamyces*, the D2 region appears to allow sufficient differentiation between the other known genera and (geno)species. The amplicon is thus well suited for screening assays, e.g., to analyze the presence, abundance, and diversity of *Neocallimastigomycetes* in interesting habitats in a fair level. This can provide valuable information to decide if it is worth examining certain samples in more detail.

Both, molecular and isolation/cultivation methods are complementary, and a polyphasic approach is the basis of the active development of taxonomic analysis. Molecular tools are designed based on known sequences. As more species are detected, cultured, and described, the available primers must be checked if the new discoveries are amplified, or if modifications are needed. In turn, molecular tools that are (re)constructed according to the latest information will deliver novel sequence information. Thus, this is a dynamic development, as databases are constantly updated, and new species are discovered.

## 4. Conclusions

Anaerobic fungi of the class *Neocallimastigomycetes* are of particular interest for biotechnology applications due to their valuable ability to unlock recalcitrant lignocellulosic materials for further utilization as renewable biofuels and for other emerging technologies. Efficient methods for their detection, such as the AGF-LSU-EnvS primer pair developed in this study, are thus highly valuable.

The main advantage of the proposed primer pair is that it is specific for the class *Neocallimastigomycetes* and that it can be used simultaneously for Illumina amplicon sequencing, phylogenetic analysis, and quantification by Real-Time PCR (qPCR) of diverse and simple *Neocallimastigomycetes* communities from environmental samples. Numerous environmental samples can first be examined by conventional PCR and further detailed by qPCR to determine the presence and abundance of *Neocallimastigomycetes*, respectively. The examination of melting curve patterns is advised, particularly for samples with a low concentration of *Neocallimastigomycetes* as well as with samples from unexplored habitats. This was demonstrated with the mara feces, where the shifted melting curve prompted amplicon sequencing, and thereby it was confirmed that the community in this host was shaped by a novel clade. Therefore, the use of the described primer set helps to rapidly identify and prioritize habitats or environmental samples worth to be further analyzed.

A continuous update to the taxonomy database and reference files will progressively complete the annotation step by optimizing the present and possibly other bioinformatics pipelines with automatic classification and clustering steps. The phylogenetic resolution of the primer pair is partially limited in-depth analyses of some closely related species, e.g., of the *Caecomyces*/*Cyllamyces*, between *Orpinomyces* clades, and to distinct strains. For these purposes, other genetic markers can be applied once an interesting environmental sample is identified. The use of long-reads, sequenced genomes, and transcriptomes can help to clear close related species or stains.

Therefore, the presented simultaneous approach will help to accelerate and simplify scanning environmental samples for *Neocallimastigomycetes* and minimize the number of samples to be examined by laborious preparations before downstream analysis. This saves time and resources, e.g., in isolation and cultivation of potential new species from environmental samples or in monitoring inoculated biocenoses in biotechnology processes. We anticipate that this tool, in combination with specific activity markers, will greatly accelerate the integration of *Neocallimastigomycetes* in green biotechnology applications.

## Figures and Tables

**Figure 1 microorganisms-10-01749-f001:**
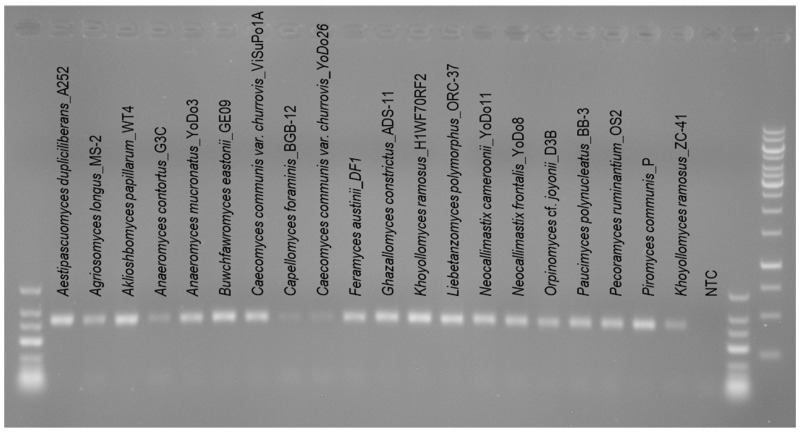
Agarose gel (1%) with amplicons of the D2 LSU subregion of 20 AGF isolates covering 18 AGF species. Amplicons, with an expected length of approximately 425 bp were generated with the AGF-LSU-EnvS-Illumina primers. GeneRuler^TM^ Low Range DNA Ladder (ThermoFisher) on the left side, second and third bands 500 bp and 400 bp, respectively. GeneRuler^TM^ 1 kb Ladder (ThermoFisher) on the far-right side, second to last bar 500 bp and last bar 250 bp. NTC = no template control.

**Figure 2 microorganisms-10-01749-f002:**
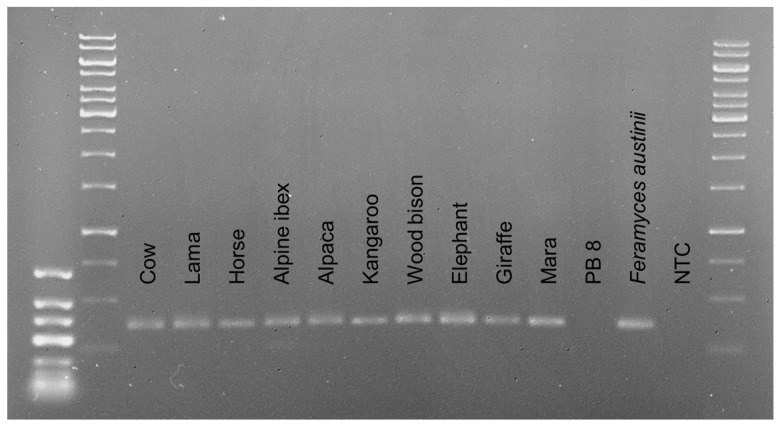
Agarose gel (1%) with amplicons of the D2 LSU subregion obtained from DNA extracts of feces of different herbivores. Amplicons generated with primer pair AGF-LSU-EnvS showed the expected length of approximately 350 bp. GeneRuler^TM^ Low Range DNA Ladder (ThermoFisher) on the left, third and fourth bands 400 bp and 300 bp, respectively. GeneRuler^TM^ 1 kb Ladder (Thermo Fisher) on the right, second to last bar 500 bp and last bar 250 bp. DNA of *Feramyces austinii* isolate DF1 was amplified as a positive control and DNA from digester sludge from a biogas plant without cattle manure input (PB 8) as a negative control. NTC = no template control.

**Figure 3 microorganisms-10-01749-f003:**
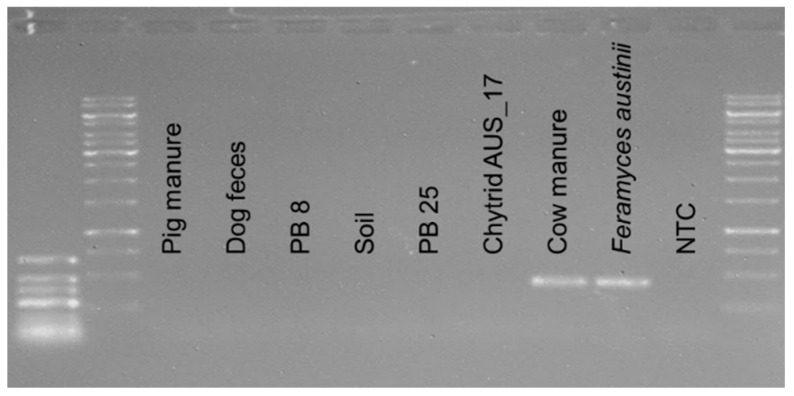
Agarose gel (1%) with amplicons of the D2 LSU subregion obtained from DNA extracts of different environmental samples (Table 2) generated with primer pair AGF-LSU-EnvS-Illumina. GeneRuler^TM^ Low Range DNA Ladder (ThermoFisher) on the left, third and fourth bands 400 bp and 300 bp, respectively. GeneRuler^TM^ 1 kb Ladder (ThermoFisher) on the right, second to last bar 500 bp and last bar 250 bp. DNA extracts of *Feramyces austinii* isolate DF1 and cow feces were used as positive controls. Further sample descriptions are listed in Table 2. NTC = no template control.

**Figure 4 microorganisms-10-01749-f004:**
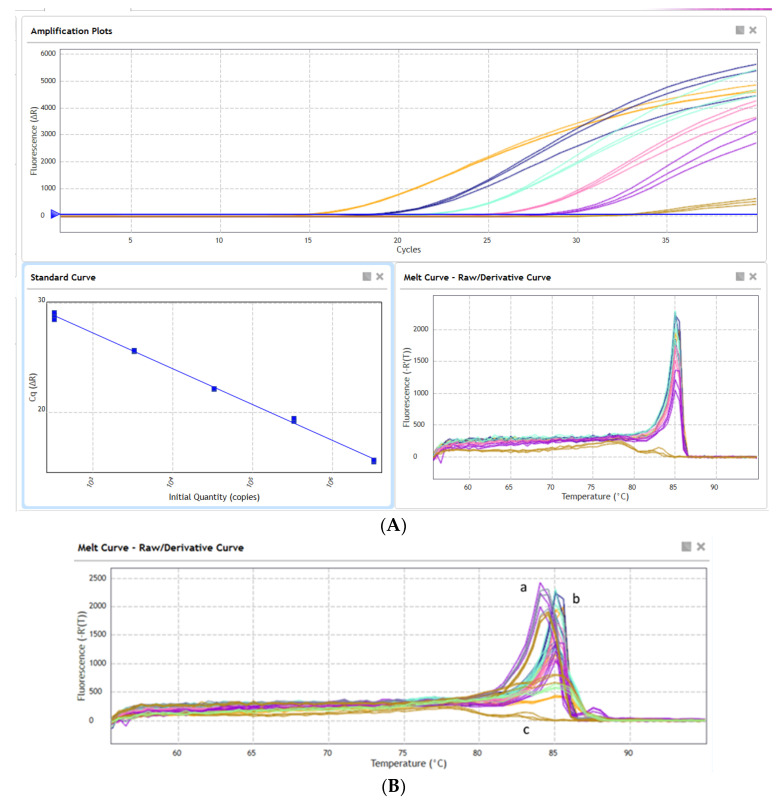
(**A**) Amplification plots and melting curves (AriaMX platform) from a *Neocallimastix cameroonii* (YoDo11) pure culture as the standard curve and NTC controls (brown color) showing no inhibition as all dilutions cross the fluorescence threshold with a difference of about 3 cycles and a melting temperature of 85.0 °C. (**B**) Melting curve plots with primer pair AGF-LSU-EnvS from DNA extracts of *Neocallimastix cameroonii* (YoDo11, standard curve), cow, elephant, lama and, mara feces. The dissociation curve generated by DNA from mara feces (a) peaked at 84.5 °C, whereas the curves of the other DNAs peaked at 85.0 °C (b), and the NTC (c, in a brown color) showed no peak.

**Figure 5 microorganisms-10-01749-f005:**
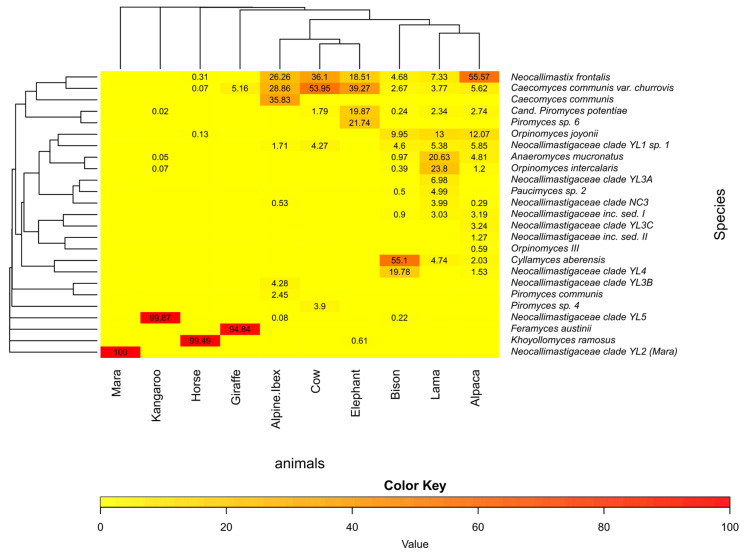
Cluster analysis heatmap visualizing the relative abundance in percentage of ASV reads per genus/species to describe the *Neocallimastigomycetes* community composition in fecal samples from ten different herbivore animals. The original table can be found as Appendix A.

**Figure 6 microorganisms-10-01749-f006:**
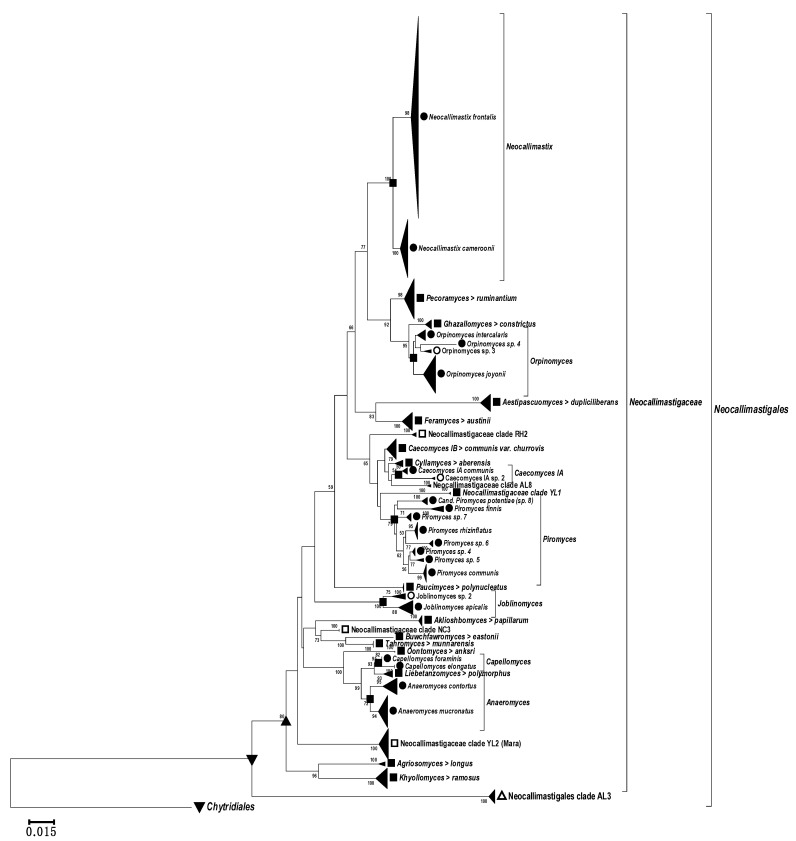
Compressed phylogenetic tree of the AGF LSU D1–D2 region. DQ536493 (*Chytriomyces* sp. WB235A) was included as an outgroup. Filled symbols: reference strain(s) of a validly published species is available; valid taxonomic assignments are in italics. Open symbols: No cultured reference strain available, species defined only by genetic traits “genospecies”. Filled triangles pointing left: compressed clade. Triangle pointing down: Order level node. Triangle pointing up: Potential family level node. Square: Genus level node (established or >95% sequence identity); “>” in assignments means that the clades have genus level but only one species can be shown. Circle: Species-level node (established or >98.5% sequence identity). The expanded phylogenetic tree is added as Appendix A.

**Figure 7 microorganisms-10-01749-f007:**
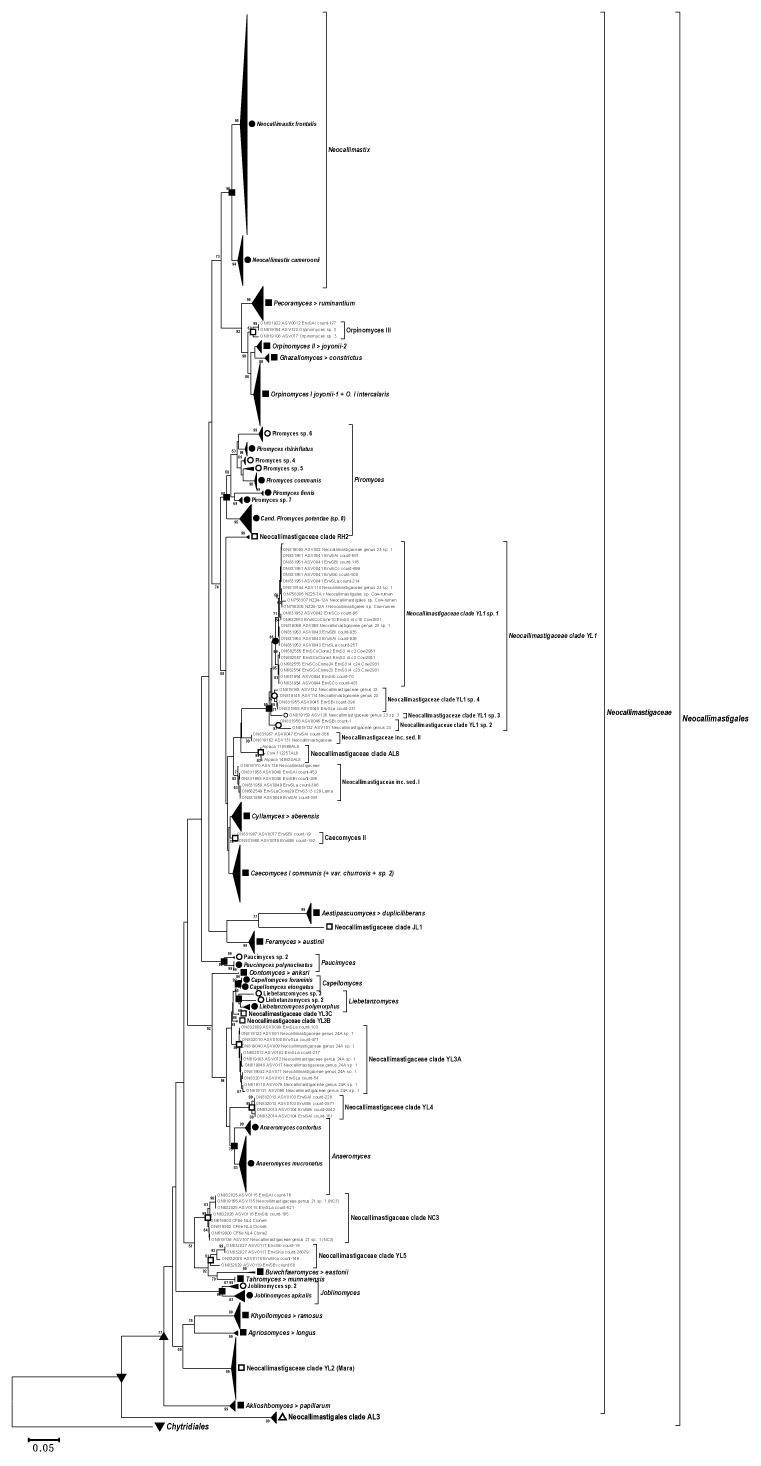
Partly compressed phylogenetic tree with the AGF-LSU-EnvS (D2 LSU) amplicons of the current study. DQ536493 (*Chytriomyces* sp. WB235A) was used as an outgroup. Clades with strong representation by Sanger and Illumina sequences are compressed, and novel clades (except for the *Neocallimastigaceae* clade YL2 “Mara clade”) are resolved. Accession numbers ON819131–ON819177 are from [25]. The expanded phylogenetic tree is added as Appendix A.

**Table 1 microorganisms-10-01749-t001:** *Neocallimastigomycetes* isolates (pure cultures) and DNA extracts used as in vitro references.

	Description	Isolate	Source	Cultivation/Extraction Reference
1	*Aestipascuomyces dupliciliberans*	A252	KIT	[24]
2	*Agriosomyces longus*	MS-2	OSU	[1]
3	*Aklioshbomyces papillarum*	WT4	OSU	[1]
4	*Anaeromyces contortus*	G3C	OSU	[1]
5	*Anaeromyces mucronatus*	YoDo3	LfL	this study
6	*Buwchfawromyces eastonii*	GE09	DoLS	this study
7	*Caecomyces communis* *var. churrovis*	ViSuPo1A	UIBK	this study
8	*Caecomyces communis* *var. churrovis*	YoDo26	LfL	this study
9	*Capellomyces foraminis*	BGB-12	OSU	[1]
10	*Feramyces austinii*	DF1	IAPG	this study
11	*Ghazallomyces constrictus*	ADS-11	OSU	[1]
12	*Khoyollomyces ramosus*	H1WF70RF2	ZHAW	[25]
13	*Khoyollomyces ramosus*	ZC-41	OSU	[1]
14	*Liebetanzomyces polymorphus*	ORC-37	OSU	[1]
15	*Neocallimastix cameroonii*	YoDo11	LfL	this study
16	*Neocallimastix frontalis*	YoDo8	LfL	this study
17	*Orpinomyces* cf. *joyonii*	D3B	OSU	[1]
18	*Paucimyces polynucleatus*	BB-3	OSU	[1]
19	*Pecoramyces ruminantium*	OS2	OSU	[1]
20	*Piromyces communis*	P	DoLS	this study
21	*Chytridiomycetes* sp.	AUS_17	OL	outgroup, this study

LfL: Bavarian State Research Center for Agriculture, Central Department for Quality Assurance and Analytics. OSU: Department of Microbiology and Molecular Genetics of the Oklahoma State University. ZHAW: Center of Environmental and Process Technology at the Zürich University of Applied Sciences. KIT: Process Engineering in Life Sciences 2: Technical Biology, at the Karlsruhe Institute of Technology. DoLS: Department of Life Sciences at Aberystwyth University. IAPG: Institute of Animal Physiology and Genetics at the Czech Academy of Sciences. OL: School of Life and Environmental Sciences, Faculty of Science at the University of Sydney

**Table 2 microorganisms-10-01749-t002:** Description of the environmental samples used to test the AGF-LSU-EnvS primer pair. HTS: high-throughput sequencing. (q)PCR: PCR based analysis, conventional or quantitative PCR.

	Sample Source	AGF Host	Origin	Characteristics	Analysis
1	Cow feces	*Bos taurus* Fleckvieh breed	Cattle farm in Dillingen, Germany	Ruminant with 4 compartments	(q)PCR, HTS, clone library
2	Lama feces	*Lama glama*	Munich Zoo, originally from South America	Ruminant with 3 compartments	(q)PCR, HTS, clone library
3	Horse feces	*Equus ferus caballus*	Private stable in Freising, Germany	Monogastric	(q)PCR, HTS
4	Alpine ibex feces	*Capra ibex ibex*	Munich zoo, originally from European alps	Ruminant with 4 compartments	(q)PCR, HTS
5	Alpaca feces	*Vicugna pacos*	Munich zoo, originally from South America	Ruminant with 3 compartments	(q)PCR, HTS
6	Kangaroo feces	*Macropus rufus*	Munich zoo, originally from Australia	Non-ruminant with 2 compartments	(q)PCR, HTS
7	Wood Bison feces	*Bison bison athabascae*	Munich zoo, originally from North America	Ruminant with 4 compartments	(q)PCR, HTS
8	Elephant feces	*Elephas maximus*	Munich zoo, originally from Asia	Monogastric	(q)PCR, HTS, clone library
9	Giraffe feces	*Giraffa camelopardalis reticulata*	Munich zoo, originally from East Africa	Ruminant with 4 compartments	(q)PCR, HTS
10	Mara feces	*Dolichotis patagonum*	Munich zoo, originally from Patagonia, South America	Monogastric, Rodent	(q)PCR, HTS, clone library
11	Pig feces	no known AGF host	Pig farm, Germany	Monogastric	(q)PCR
12	Dog feces	no known AGF host	Pet, Freising, Germany	Monogastric	(q)PCR
13	Digestate from fermenter of agricultural biogas plant PB 8	Substrates from no AGF host	Operating biogas plant, Germany	Feedstock only of energy plants (triticale, grass, maize silage)	(q)PCR
14	Digestate from fermenter of agricultural biogas plant PB 25	36% of the substrates comes from an AGF host	Operating biogas plant, Germany	biogas plant fed with 36% cow manure	(q)PCR
15	Soil	no AGF host	Freising, Germany	Grassland, not of agricultural use	(q)PCR

**Table 3 microorganisms-10-01749-t003:** Characteristics of the primer pair AGF-LSU-EnvS developed to detect, quantify, and classify *Neocallimastigomycetes* in environmental samples.

Primer Name	Sequence (5′→3′)	Length (Bases)	T_m_ (°C) *	Amplicon Size (bp)	Target
AGF-LSU-EnvS F	GCGTTTRRCACCASTGTTGTT	21	59.8	349–355	D2 region of the 28S rRNA coding gene (LSU)
AGF-LSU-EnvS R	GTCAACATCCTAAGYGTAGGTA	22	58.4		

* T_m_ = 69.3 + 0.41 * A—650/B, where A = C+G content in %; and B = nucleotide length of the primer [40].

**Table 4 microorganisms-10-01749-t004:** Composition of mock communities prepared of pooled LSU D2 Illumina amplicons from different AGF isolates, and analysis by the DADA2-QIIME2 bioinformatics pipeline. Mock community 1: 17 isolates in a 5.3% ratio each, and *Caecomyces communis* var. *churrovis* 10.5%. Mock community 2: *Khoyollomyces ramosus* (90%) and *Agriosomyces longus* (10%). * 2 ASVs allocated as *P. communis* (275 and 154 reads) and 1 ASV as *Piromyces* sp. 6. (26 reads).

	Mock Community 1				Mock Community 2			
Isolate	Reads per Isolate	%	% Pooled	Number of ASVs	Reads per Isolate	%	% Pooled	Number of ASVs
*Aestipascuomyces dupliciliberans* A252	254	3.2%	5.3%	4			-	
*Agriosomyces longus* MS-2	410	5.2%	5.3%	2	608	8.4%	10.0%	2
*Aklioshbomyces papillarum* WT-4	371	4.7%	5.3%	2			-	
*Anaeromyces contortus* G3C	394	5.0%	5.3%	5	8	0.1%	-	1
*Anaeromyces mucronatus* YoDo3	290	3.7%	5.3%	2			-	
*Buwchfawromyces eastonii* GE09	460	5.9%	5.3%	2			-	
*Caecomyces communis* var. *churrovis* ViSoPu1A	837	10.7%	10.5%	2			-	
*Capellomyces foraminis* BGB-12	255	3.3%	5.3%	1			-	
*Feramyces austinii* DF1	391	5.0%	5.3%	3			-	
*Ghazallomyces constrictus* ADS-11	433	5.5%	5.3%	2			-	
*Khoyollomyces ramosus* H1WF70RF2	485	6.2%	5.3%	2	6560	90.8%	90.0%	2
*Liebetanzomyces polymorphus* ORC-37	419	5.3%	5.3%	1			-	
*Neocallimastix cameroonii* YoDo11	448	5.7%	5.3%	4			-	
*Neocallimastix frontalis* YoDo8	515	6.6%	5.3%	4			-	
*Orpinomyces* cf. *joyonii* D3B	449	5.7%	5.3%	4			-	
*Paucimyces polynucleatus* BB-3	537	6.9%	5.3%	1			-	
*Pecoramyces ruminantium* OS2	434	5.5%	5.3%	2			-	
*Piromyces communis* P	455	5.8%	5.3%	3 *	42	0.6%	-	1
Not assigned	0	0.0%	-	0	3	0.04%	-	1
Total	7837	100%	100%		7221	100%	100%	

## Data Availability

The Sanger sequences to complement the LSU D1–D2 region reference files were registered under the accession numbers ON614226–ON614635, ON619892–ON619903, ON650566–ON650603, ON758302–ON758335, ON695782–ON6795796, and ON695797–ON695831. The Sanger sequences from the clone libraries (LSU D2 region) were deposited in the NCBI GenBank database by the accession numbers ON682515 to ON682549 and ON682552 to ON682686. The Illumina HTS data were deposited in the NCBI GenBank database (SRA accession numbers SRR19593063 to SRR19593072 from the BioProject PRJNA846766, as well as derived ASVs with accession numbers ON831911–ON832040). The alignments (D1–D2 LSU and D2 LSU) can be obtained by the authors upon request.

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
