# Peer review of "Simultaneous Metabarcoding and Quantification of Neocallimastigomycetes from Environmental Samples: Insights into Community Composition and Novel Lineages"

_microorganisms, 2022, doi:10.3390/microorganisms10091749_

Round 1

Reviewer 1 Report

I was excited to see this research paper as a well designed validated priemr set to conduct amplicon studies targeting anaerobic fungi are needed. The authors describe research aimed at developing and validating a novel primer set for amplicon studies targeting anaerobic fungi. This is an important advancement in the field of rumen microbiology and mycology because the currently available primers do not adequately cover the taxonomic diversity of the phylum Neocallimastigomycota. The authors have done a very thorough and in depth study to design a novel primer set, validate the primers and test them using environmental samples. This is a very well done study and the paper is written clearly. I have no concerns about the methods used or the interpretation of the results. This is a very well done contribution to the field. 

Author Response

Dear Reviewer 1, thank you for your words of praise and for emphasizing the relevance of this study. We appreciate that you invested your time to review this paper. We hope that this tool will be useful for many researchers.

Reviewer 2 Report

The manuscript on Simultaneous metabarcoding and quantification of Neocallimastigomycetes from environmental samples: insights into community composition and novel lineages by Young et al. describes an approach to uncovering the diversity of symbiotic anaerobic fungi of ruminants. The fungi are important from an agricultural and biotechnological perspective therefore the subject is of wide interest. Due to problems posed by the group regarding their cultivation and classification based on traditional methods the molecular approach is the method of choice. The ITS-LSU sequence is not perfect for taxonomy classification but so far the best choice (or even the only one available).

Authors of the manuscript describe their designed primers as superior to those published previously by other groups. However, they did not provide experimental (wet lab) data generated with the other for comparison with their own.

The primers allow amplifying the tested 18 species (Fig 1) however with significantly different efficiency (clearly visible in results presented in Table 4) that need to be clearly stated/elaborated since it influences quantification assays. If available authors should provide amplification efficiencies calculated from standard curve slopes prepared on different species. A standard curve prepared on specific species cannot be extrapolated/used on/for different species (the paragraph in lines 490-497 needs thorough revision). 

Specific comments:

(1) lines 88-90 the sentence "Some of the amplicons are also too long (>600 bp) to be useful in (...) high throughput sequencing (HTS)" - consider changing since the improvement of long-read NGS.

(2) provide a reference for the statement that "(...) D1-D2 region has been lately more accepted..." (lines 101-103).

(3) please clearly indicate the names of the two species used for the formulation of Mock community 2 (lines 292-293) in the Methods section.

(4) Table 3 provides primer pair AGF-LSU-EnvS details - please correct the table as both primers generate amplicon of size 349-355 bp (not single primer - separate amplicon).

(5) Figure 4.2 - correct placements of 'a', 'b', and 'c' on the graph and include the color name for the NTC curve (brown?) in the description.

(6) Consider changing the expression 'normalized percentage' into 'relative abundance' in the description of NGS results.

The manuscript is well organized, clearly written, and easy to understand. There are only minor drawbacks. 

Author Response

Dear Reviewer 2,

thank you for taking the time to review our article. We appreciate your input and comments.

Attached you will find a PDF with the detailed answers to each of your comments and suggestions.

Please let us know if we managed to answer your inquiries.
